# T-Cell Responses to COVID-19 Vaccines and Breakthrough Infection in People Living with HIV Receiving Antiretroviral Therapy

**DOI:** 10.3390/v16050661

**Published:** 2024-04-24

**Authors:** Sneha Datwani, Rebecca Kalikawe, Rachel Waterworth, Francis M. Mwimanzi, Richard Liang, Yurou Sang, Hope R. Lapointe, Peter K. Cheung, Fredrick Harrison Omondi, Maggie C. Duncan, Evan Barad, Sarah Speckmaier, Nadia Moran-Garcia, Mari L. DeMarco, Malcolm Hedgcock, Cecilia T. Costiniuk, Mark Hull, Marianne Harris, Marc G. Romney, Julio S. G. Montaner, Zabrina L. Brumme, Mark A. Brockman

**Affiliations:** 1Faculty of Health Sciences, Simon Fraser University, Burnaby, BC V6A 1S6, Canada; sneha_datwani@sfu.ca (S.D.); rebecca_kalikawe@sfu.ca (R.K.); rachel_waterworth@sfu.ca (R.W.); francis_method_mwimanzi@sfu.ca (F.M.M.); yurou_sang@sfu.ca (Y.S.); pcheung@bccfe.ca (P.K.C.); fredrick_omondi@sfu.ca (F.H.O.); maggie_duncan@sfu.ca (M.C.D.); evan_barad@sfu.ca (E.B.); 2British Columbia Centre for Excellence in HIV/AIDS, Vancouver, BC V6Z 1Y6, Canada; rliang@bccfe.ca (R.L.); hrlapointe@bccfe.ca (H.R.L.); nmorangarcia@bccfe.ca (N.M.-G.); mhull@bccfe.ca (M.H.); mharris@bccfe.ca (M.H.); drjm@bccfe.ca (J.S.G.M.); 3Department of Pathology and Laboratory Medicine, Providence Health Care, Vancouver, BC V6Z 1Y6, Canadamromney@providencehealth.bc.ca (M.G.R.); 4Department of Pathology and Laboratory Medicine, University of British Columbia, Vancouver, BC V6T 1Z4, Canada; 5Spectrum Health, Vancouver, BC V6Z 2T1, Canada; malcolmh@spectrum-health.net; 6Division of Infectious Diseases Chronic Viral Illness Service, McGill University Health Centre, Montreal, QC H4A 3J1, Canada; cecilia.costiniuk@mcgill.ca; 7Research Institute of the McGill University Health Centre, Montreal, QC H4A 3J1, Canada; 8Department of Medicine, University of British Columbia, Vancouver, BC V6T 1Z4, Canada; 9Department of Family Practice, Faculty of Medicine, University of British Columbia, Vancouver, BC V6T 1Z4, Canada; 10Department of Molecular Biology and Biochemistry, Simon Fraser University, Burnaby, BC V6A 1S6, Canada

**Keywords:** COVID-19, vaccine, SARS-CoV-2, coronavirus, T cells, HIV, antiretroviral therapy

## Abstract

People living with HIV (PLWH) can exhibit impaired immune responses to vaccines. Accumulating evidence indicates that PLWH, particularly those receiving antiretroviral therapy, mount strong antibody responses to COVID-19 vaccines, but fewer studies have examined cellular immune responses to the vaccinations. Here, we used an activation-induced marker (AIM) assay to quantify SARS-CoV-2 spike-specific CD4+ and CD8+ T cells generated by two and three doses of COVID-19 vaccines in 50 PLWH receiving antiretroviral therapy, compared to 87 control participants without HIV. In a subset of PLWH, T-cell responses were also assessed after post-vaccine breakthrough infections and/or receipt of a fourth vaccine dose. All participants remained SARS-CoV-2 infection-naive until at least one month after their third vaccine dose. SARS-CoV-2 infection was determined by seroconversion to a Nucleocapsid (N) antigen, which occurred in 21 PLWH and 38 control participants after the third vaccine dose. Multivariable regression analyses were used to investigate the relationships between sociodemographic, health- and vaccine-related variables, vaccine-induced T-cell responses, and breakthrough infection risk. We observed that a third vaccine dose boosted spike-specific CD4+ and CD8+ T-cell frequencies significantly above those measured after the second dose (all *p* < 0.0001). Median T-cell frequencies did not differ between PLWH and controls after the second dose (*p* > 0.1), but CD8+ T-cell responses were modestly lower in PLWH after the third dose (*p* = 0.02), an observation that remained significant after adjusting for sociodemographic, health- and vaccine-related variables (*p* = 0.045). In PLWH who experienced a breakthrough infection, median T-cell frequencies increased even higher than those observed after three vaccine doses (*p* < 0.03), and CD8+ T-cell responses in this group remained higher even after a fourth vaccine dose (*p* = 0.03). In multivariable analyses, the only factor associated with an increased breakthrough infection risk was younger age, which is consistent with the rapid increase in SARS-CoV-2 seropositivity that was seen among younger adults in Canada after the initial appearance of the Omicron variant. These results indicate that PLWH receiving antiretroviral therapy mount strong T-cell responses to COVID-19 vaccines that can be enhanced by booster doses or breakthrough infection.

## 1. Introduction

Before the widespread availability of COVID-19 vaccines, there was evidence that people living with HIV (PLWH) were at increased risk of severe COVID-19 outcomes [1,2,3,4,5], making vaccination particularly important in this group. Since then, evidence has indicated that COVID-19 vaccines effectively prevent severe disease associated with SARS-CoV-2 infection, including in PLWH [6,7]. Studies analyzing vaccine-induced immune responses, including from our group, also confirm that PLWH, particularly those receiving suppressive antiretroviral therapy, mount robust humoral (antibody) responses to COVID-19 immunization [8,9,10,11,12,13,14,15]. However, comparably fewer studies have investigated the cellular immune response to COVID-19 vaccines in PLWH, which includes CD4+ helper T cells that play a central role in generating antigen-specific B cells and antibodies and CD8+ cytotoxic T cells that recognize and eliminate virus-infected cells [16,17,18,19]. Indeed, while COVID-19 vaccines typically induce potent T-cell responses in PLWH [18,19,20,21,22,23], the frequency of spike-specific CD4+ T cells following COVID-19 vaccination may be lower in PLWH, particularly among those with low CD4+ T-cell counts [24,25,26,27]. A better understanding of T-cell responses elicited by COVID-19 vaccines and breakthrough SARS-CoV-2 infections will inform ongoing efforts to enhance protective immunity in PLWH.

Towards this goal, we investigated the dynamics of spike-specific CD4+ and CD8+ T-cell responses elicited after two and three doses of COVID-19 vaccine in a cohort of 50 adult PLWH receiving antiretroviral therapy and 87 control participants without HIV. All participants remained SARS-CoV-2 infection-naive until at least one month after their third vaccine dose. We also investigated spike-specific CD4+ and CD8+ T-cell responses in a subset of 21 PLWH who experienced their first SARS-CoV-2 breakthrough infection between one and six months after their third vaccine dose. Overall, our results indicated that vaccine-induced spike-specific CD4+ and CD8+ T-cell frequencies were boosted with each vaccine dose and that these responses were broadly comparable between PLWH and controls, except for CD8+ T-cell responses after the third dose, which were moderately lower in PLWH compared to controls.

## 2. Methods

Participants. Our cohort of COVID-19 vaccine recipients, based in British Columbia (BC), Canada, has been described previously [9,12,28]. Of the 99 PLWH and 152 controls (without HIV) initially enrolled in the study, 62 (PLWH) and 117 (controls) remained naive for SARS-CoV-2 infection until at least one month after their third COVID-19 vaccine dose. We examined a subset of 50 PLWH and 87 controls, selected primarily based on sample availability (Table 1).

Ethics approval. All participants provided written informed consent. The University of British Columbia/Providence Health Care and Simon Fraser University Research Ethics Boards approved this study (protocol H20-03906, initial approval date 15 December 2020; and H21-00742, initial approval date 24 March 2021).

SARS-CoV-2 seroconversion. SARS-CoV-2 infection was identified based on the development of serum antibodies against the viral Nucleocapsid (N) protein using the Elecsys Anti-SARS-CoV-2 assay (Roche Diagnostics, Mississauga, ON Canada), combined with diagnostic (PCR or rapid antigen test) information where available.

T-cell-activation-induced marker assays. Cryopreserved peripheral blood mononuclear cells (PBMC) were thawed and diluted in TexMACS media (Miltenyi Biotec, Auburn, CA, USA; Cat#130-097-196). PBMC were stimulated at 1 × 10^6^ cells per well for 24 h with peptide pools spanning the SARS-CoV-2 ancestral spike protein (consisting of 315 mainly 15-mer peptides, overlapping by 11 amino acids) (Miltenyi Biotec, Cat#130-127-953) in duplicate in a 96-well U-bottom plate. PBMC were incubated with media only (no peptide) as a negative control or two µg/mL of a CytoStim reagent (Miltenyi Biotec, Cat#130-092-172) as a positive control. Following stimulation, the cells were labeled with CD8-APC/Cyanine7 (Biolegend, Cat#301016), CD4-FITC (Biolegend, San Diego, CA, USA; Cat#300538), CD137-APC (Biolegend, Cat#309810), CD69-PE (BD, Cat#555531), OX40-PE-Cy7 (Biolegend, Cat#350012), CD3-PerCP/Cyanine5.5 (Biolegend, Cat#317336), CD14-V500 (BD Bioscinces, Mississauga, ON Canada Cat#561391), CD19-V500 (BD Biosciences, Cat#561121), and a 7-AAD Viability Staining Solution (Biolegend, Cat#420404). Data were acquired on a Beckman Coulter Cytoflex S flow cytometer, with a minimum of 10,000 CD3+ T cells assayed per participant. After identifying CD3+CD4+ and CD3+CD8+ T-cell subsets, the percentage of stimulated cells was determined based on the upregulation of two activation markers, using CD137 and OX40 for CD4+ T cells and CD137 and CD69 for CD8+ T cells (see gating strategy in Appendix A). Data were analyzed in FlowJo version 10.8.1 (BD Biosciences).

SARS-CoV-2 neutralization. We previously reported the ability of participant plasma to neutralize ancestral SARS-CoV-2 (isolate USA-WA1/2020; BEI Resources, Manassas, VA, USA; Cat #NR-52281) based on the results of a live-virus assay [28].

Statistical analyses. Continuous variables were compared using the Mann–Whitney U-test (unpaired data) or the Wilcoxon test (paired measures). Relationships between continuous variables were assessed using Spearman’s correlation. Zero-inflated beta regressions were used to investigate the relationship between sociodemographic factors and vaccine-induced T-cell responses using a confounder model that adjusted for variables that could either influence these responses or that differed in prevalence between groups. These regressions model the response variable as a beta-distributed random variable whose mean is given by a linear combination of the predictor variables (after a logit transformation). Beta distributions are bounded below and above by 0 and 100%, making this a standard choice of regression for frequency data. Because a beta distribution does not admit values of 0 (or 100), we used a zero-inflated beta distribution, which allows for zeros in the data, to accommodate the minority of non-responders (i.e., 0 values) in our data. Analyses performed after two-dose vaccination included the following variables: HIV status (non-PLWH as reference), age (per year), sex at birth (female as reference), ethnicity (non-white as reference), chronic conditions (per condition), a ChAdOx1-containing initial vaccine regimen (mRNA only vaccination as reference), and the interval between first and second doses (per day). Analyses performed after three-dose vaccination additionally included the third COVID-19 mRNA dose brand (BNT162b2 as reference), the interval between the second and third doses (per day), and the % spike-specific T cells after two doses (per percent increase). Multivariable logistic regression was used to explore the relationship between these variables and the risk of a SARS-CoV-2 breakthrough infection between one and six months after the third vaccine dose. All tests were two-tailed, with *p* < 0.05 considered statistically significant. Analyses were conducted using Prism v9.2.0 (GraphPad, Boston, MA, USA) and in R v4.3.2.

## 3. Results

### 3.1. Participant Details

Characteristics of the 50 PLWH and 87 control participants without HIV are shown in Table 1. All participants remained SARS-CoV-2 infection-naive until at least one month after their third vaccine dose, as confirmed by repeated negative serology for SARS-CoV-2 anti-N antibodies and a lack of reporting of any SARS-CoV-2-positive test results (either by PCR or rapid antigen test) up to this time point. PLWH and controls were a median age of 58 (interquartile range [IQR] 42–65) and 50 (IQR 35–72) years old, respectively, and predominantly of white ethnicity (72% of PLWH and 57% of controls). Most PLWH were male (88%, reflecting the overall population of PLWH in BC). In comparison, most controls were female (67%, broadly reflecting the overall demographics of healthcare workers in the region, who were predominantly recruited as controls). Both groups reported few chronic health conditions (excluding HIV infection; median 1 [IQR 0–1] in PLWH, versus median 0 [IQR 0–1] in controls). All PLWH were receiving combination antiretroviral therapy, with a median HIV plasma viral load of <50 copies/mL at study entry. At study entry, the median CD4+ T-cell count among the PLWH was 695 (IQR 468–983) cells per mm^3^, while the median nadir CD4+ T-cell count was 250 (IQR 140–490) cells per mm^3^.

Most participants (84% of PLWH and 97% of controls) initially received two doses of an mRNA vaccine, either BNT162b2 or mRNA-1273; the remainder received at least one ChAdOx1 dose in their initial series. Among PLWH, second-vaccine doses were administered at a median of 59 days after the first dose, compared to a median interval of 89 days among controls (*p* < 0.0001). This is because Canada delayed second doses well beyond the manufacturers’ recommended 21–28 day interval due to initially limited availability of vaccine supplies in the country [29]. As a result, individuals who received their first doses early in the vaccination campaign, including most healthcare workers, generally waited the longest to receive their second doses. Retrospectively, it was determined that delaying second doses did not reduce vaccine efficacy and, in some cases, may have enhanced protection and reduced the risk of myocarditis, prompting the World Health Organization to recommend a 2-month (~60-day) interval between the first and second doses [30]. As a result, our multivariable analyses adjusted for between-dose intervals and other sociodemographic and clinical attributes listed in Table 1. Third vaccine doses were predominantly mRNA-1273 (66% of PLWH and 61% of controls) and were administered at a median of 182 to 197 days (approximately 6 to 6.5 months) after the second dose. Third mRNA-1273 doses also differed by age and health history: as per vaccine guidelines in British Columbia, adults 65 years of age and older, as well as PLWH with a history of immunodeficiency, were eligible for a 100 µg dose, whereas all others were eligible for a 50 µg booster dose. All BNT162b2 third doses were 30 µg, which is equivalent to the first and second doses. A total of 21 (42%) of PLWH and 38 (44%) of the controls experienced their first SARS-CoV-2 infection between one and six months after the third vaccine dose, as determined by seroconversion to the SARS-CoV-2 N protein and self-reported positive SARS-CoV-2 test results where available. Though viral genotyping was not performed for these “breakthrough” infections, these were likely Omicron BA.1 or BA.2 based on local molecular epidemiology trends [31].

### 3.2. T-Cell Responses following Two and Three COVID-19 Vaccine Doses

Before vaccination, the percentage of spike-specific CD4+ and CD8+ T cells, measured in a subset of 10 PLWH and 15 controls, was negligible, as expected (Figure 1A,B). Following two vaccine doses, the percentage of spike-specific CD4+ T cells increased significantly from pre-vaccine levels (paired measures *p* ≤ 0.004 for both PLWH and controls), reaching a median 0.25% (IQR 0.07–0.50) among PLWH and a median 0.32% (IQR 0.15–0.55) among controls, a difference that was not statistically significant between the groups (*p* = 0.11; Figure 1A). CD8+ T-cell responses also increased significantly from pre-vaccine levels following two vaccine doses (paired measures *p* ≤ 0.002 in both groups), reaching a median of 0.39% (IQR 0.22–0.53) among PLWH and a median of 0.31% (IQR 0.14–0.61) among controls, a difference that was not statistically significant between the groups (*p* = 0.39; Figure 1B). The lack of statistically significant differences in vaccine-induced T-cell responses between PLWH and controls remained after controlling for sociodemographic and health- and vaccine-related variables (*p* = 0.08 for CD4+ T-cell responses; *p* = 0.99 for CD8+ T-cell responses; Appendix A). Of note, these multivariable analyses did not identify any variables that were significantly associated with spike-specific CD4+ T-cell responses after two vaccine doses (Appendix A), though having received at least one ChAdOx1 dose as part of the initial vaccine series was weakly associated with higher spike-specific CD8+ T-cell responses after the second vaccine dose (*p* = 0.049; Appendix A).

A third vaccine dose further boosted the percentage of spike-specific CD4+ T cells beyond the levels observed after two doses (paired measures *p* < 0.0001 for both PLWH and controls; Figure 1A). After the third dose, spike-specific CD4+ T-cell frequencies reached a median of 0.67% (IQR 0.50–0.86) among PLWH compared to a median of 0.71% (IQR 0.40–0.98) among controls (*p* = 0.52; Figure 1A), where this lack of significant difference between the groups remained after controlling for sociodemographic and health- and vaccine-related variables (*p* = 0.58; Appendix A). Similarly, a third vaccine dose further boosted the percentage of spike-specific CD8+ T cells (paired measures *p* < 0.0001 for both groups; Figure 1B). Somewhat in contrast to the other immune measures, however, the median CD8+ T-cell response among PLWH (0.61%; IQR 0.43–0.79) was lower compared to that among controls (0.78%; IQR 0.43–1.1) after the third vaccine dose (*p* = 0.02; Figure 1B). This difference remained marginally statistically significant after controlling for sociodemographic and health- and vaccine-related variables (*p* = 0.045; Appendix A) and may reflect the persisting exhaustion of CD8+ T cells that has been observed in some PLWH.

The multivariable analyses also revealed that the strongest predictors of spike-specific CD4+ and CD8+ T-cell responses after three vaccine doses were the corresponding percentage of spike-specific T cells achieved after two doses (Appendix A). For example, the zero-inflated beta regression estimates for the impact of each 1% increment in post-second-dose T-cell frequencies on post-third-dose frequencies were 0.71 for CD4+ T cells (*p* < 2 × 10^−16^) and 0.79 for CD8+ T cells (*p* < 2 × 10^−16^). Male sex, receipt of an mRNA-1273 booster, and, somewhat unexpectedly, a higher number of chronic health conditions were also independently associated with higher spike-specific CD4+ T-cell responses (estimates 0.11–0.24; all *p* < 0.04; Appendix A). We hypothesize that the association between chronic health conditions and better T-cell responses is because, after adjusting for post-second-dose responses, individuals with such conditions benefited particularly from a third dose. The association with mRNA-1273 is likely because adults 65 years and older and PLWH with a history of immune dysfunction were eligible to receive the higher (100 µg) mRNA-1273 third dose in British Columbia. In contrast, the general population received the standard 50 µg booster dose.

### 3.3. Correlations between Vaccine-Induced T-Cell Responses and Other Immune Measures

The percentages of spike-specific CD4+ and CD8+ T cells were strongly associated with one another after the second vaccine dose (Spearman’s ρ = 0.57; *p* < 0.0001) as well as after the third dose (Spearman’s ρ = 0.54; *p* < 0.0001; Figure 2A). No association was observed between the percentage of spike-specific CD4+ T cells and the magnitude of vaccine-induced neutralizing antibody responses based on a live-virus assay (reported previously by our group [28]) after either the second (Spearman’s ρ = −0.04, *p* = 0.7) or third vaccine dose (Spearman’s ρ = −0.07, *p* = 0.4; Figure 2B). Prior studies have linked the frequency of CD4+ T follicular helper cells with antibody responses [32], but we did not assess T-cell subsets here.

As CD4+ T-cell counts are used to indicate immune functions in PLWH, we evaluated the relationship between this parameter and vaccine-induced T-cell responses in this group. We observed no association between the most recent CD4+ T-cell count and the frequency of either spike-specific CD4+ or CD8+ T cells following the second or third vaccine doses (Spearman’s ρ = −0.09 to −0.18; all *p* > 0.2; Appendix A). Similarly, we observed no association between nadir CD4+ T-cell counts and vaccine-induced CD4+ T-cell responses after either dose (Spearman’s ρ = −0.05 and −0.12, respectively, both *p* > 0.4) (Appendix A). By contrast, an unexpected modest negative correlation was found between nadir CD4+ T-cell counts and spike-specific CD8+ T-cell responses after both the second (Spearman’s ρ = −0.29, *p* = 0.04) and third (Spearman’s ρ = −0.28, *p* = 0.05) dose. We hypothesize that elevated CD8+ T-cell responses are, in part, because PLWH with a history of immune dysfunction were eligible to receive a higher (100 µg) third dose of mRNA-1273, though this would not explain our observations after the second dose. Regardless, there is no indication in our data that a low nadir CD4+ T-cell count impaired cellular responses to COVID-19 vaccination in our cohort.

### 3.4. Breakthrough Infection Further Boosts CD4+ and CD8+ T-Cell Responses

In British Columbia, the initial waves of Omicron BA.1 and BA.2 occurred after the mass administration of third vaccine doses in the province. Of the 50 PLWH studied, 21 (42%) experienced SARS-CoV-2 breakthrough infections between 1 and 6 months following their third dose. Post-breakthrough plasma and PBMC specimens were available from nearly all PLWH, so we further examined humoral (neutralizing antibody) and cellular immune responses six months after the third dose and again one month after receipt of a fourth vaccine dose in the PLWH cohort (*n* = 50) (Figure 3). As we reported previously [11,28], neutralizing antibody activity in plasma declined significantly among SARS-CoV-2-naive individuals after the third vaccine dose. Still, in those who experienced a breakthrough infection, neutralization activity increased to above-peak levels induced by vaccination. For example, at six months post-third-vaccine dose, the median virus neutralization titer (reported as a reciprocal dilution) was 640 (IQR 320–2560) in PLWH who experienced a breakthrough infection, double the observed value in this group after three vaccine doses alone (paired test, *p* = 0.002; Figure 3A). This neutralization titer was also eight times higher than the median titer of 80 (IQR 20–160) observed at six months post-third-vaccine dose in PLWH who remained SARS-CoV-2-naive (*p* < 0.0001; Figure 3A). The observation that neutralization titers increase significantly following a breakthrough infection confirms the ongoing presence of spike-specific memory B cells despite diminished neutralizing activity in the blood.

In contrast to neutralizing antibodies, spike-specific CD4+ and CD8+ T-cell frequencies did not wane in SARS-CoV-2-naive PLWH following vaccination. The percentage of both spike-specific CD4+ and CD8+ T cells increased slightly between 1 and 6 months after the third vaccine dose (paired *p* = 0.003 and *p* = 0.005, respectively, Figure 3B and Figure 3C), which is consistent with the continued presence of vaccine-induced T cells even in the absence of an antigen. Like humoral responses, a breakthrough infection boosted T-cell responses to levels higher than those observed in the same participants after vaccination alone (both paired tests *p* ≤ 0.0001; Figure 3B,C). Moreover, the median percentage of spike-specific CD4+ T cells at six months post-third-vaccine dose in breakthrough cases was 1.01% (IQR 0.78–1.51), compared to 0.70% (IQR 0.60–1.02) in SARS-CoV-2-naive participants at this same time point (*p* = 0.03; Figure 3B). The median percentage of spike-specific CD8+ T cells in breakthrough cases was 1.20% (IQR 0.96–1.55), compared to 0.80% (IQR 0.60–1.02) in SARS-CoV-2-naive participants at this time point (*p* = 0.002; Figure 3C).

A fourth vaccine dose significantly increased neutralizing antibody responses and spike-specific T-cell responses in SARS-CoV-2-naive PLWH (all paired tests *p* ≤ 0.0001; Figure 3A–C), illustrating the immune benefit of a fourth vaccine dose in the absence of a SARS-CoV-2 infection. A fourth dose also significantly increased humoral and cellular immune responses in PLWH who experienced prior breakthrough infections (all paired tests *p* < 0.03; Figure 3A–C). Of note, post-fourth-dose neutralizing antibody titers and CD8+ T-cell responses in PLWH who experienced a breakthrough infection were significantly higher than among SARS-CoV-2-naive PLWH at this same time (*p* = 0.001 and *p* = 0.03, respectively; Figure 3A and Figure 3C). This is consistent with prior reports of superior “hybrid” immunity generated by a combination of vaccinations and infections [33,34].

### 3.5. Correlates of Protection against Breakthrough Infection

Given the substantial proportion of participants who experienced their first SARS-CoV-2 breakthrough infection between one and six months after receiving three vaccine doses (42% of PLWH; 44% of controls), we explored our data for potential correlates of protection. As HIV status was not associated with breakthrough infection (*p* = 0.86), PLWH and control groups were combined for analysis, yielding 78 participants who remained SARS-CoV-2-naive and 59 participants who experienced SARS-CoV-2 breakthrough infection during the follow-up period. Univariable analyses revealed older age to be significantly associated with protection against breakthrough infection: participants remaining SARS-CoV-2-naive were a median of 60 [IQR 42–71] years old, while those who experienced a breakthrough infection were a median of 43 [IQR 34–60] years old, (combined cohort *p* = 0.001; Figure 4A). This observation was driven mainly by the control group, which included 29 adults aged 70 years or over (compared to three in the PLWH group). The percentage of spike-specific CD4+ T cells one month after the third vaccine dose was not associated with breakthrough infection (*p* = 0.72 for the combined cohort; Figure 4B), but a higher percentage of spike-specific CD8+ T cells at this time point was associated with remaining SARS-CoV-2-naive (*p* = 0.046 for the combined cohort) (Figure 4C). This observation was driven by differences in CD8+ T-cell responses among PLWH (*p* = 0.02) and suggested that cytotoxic T cells specific for SARS-CoV-2 may contribute to protection. Somewhat surprisingly, we found no association between neutralizing antibody responses induced by three vaccine doses and breakthrough infections (*p* = 0.33 for the combined cohort) (Figure 4D). This is likely because vaccine-induced antibodies against the ancestral (Wuhan) spike had limited activity against the Omicron strains that caused the majority of breakthrough infections, and because these infections occurred a median of 3.7 months after booster vaccinations when circulating antibody titers had declined substantially. In a multivariable logistic regression analysis incorporating participant sociodemographic, vaccine, and immune-response variables, older age was the only significant predictor of protection against SARS-CoV-2 breakthrough infection (Appendix A). Though initially surprising, this observation is consistent with the rapid rise in SARS-CoV-2 anti-N seroprevalence in Canada among younger individuals following the emergence of Omicron. For example, by mid-June 2022, anti-N seroprevalence in Canadian adults aged <25 years was 57%; in those aged 25–39 years, it was 51%; in those aged 40–59 years, it was 40%; and in those 60 years or older it was 25% [35]. Similar age-specific trends were observed in Finland [36] and the USA [37], suggesting that many older adults continued to benefit from efforts to limit exposure to COVID-19 in the community, even after the arrival of Omicron. While we speculate that this outcome likely reflects the impact of social and/or behavioral factors rather than biological factors on infection risk, we cannot rule out other variables that were not assessed in our study, including differences in T-cell phenotypes, host immunogenetics, and local epidemiology of circulating SARS-CoV-2 variants.

## 4. Discussion

We observed that the frequencies of SARS-CoV-2 spike-specific CD4+ and CD8+ T cells were not significantly different between PLWH and controls after two doses of a COVID-19 vaccine. CD4+ and CD8+ T-cell responses were enhanced following a third vaccine dose, but spike-specific CD8+ T-cell responses were modestly lower in PLWH compared to controls, even after adjustment for sociodemographic, health- and vaccine-related variables (*p* = 0.045). Nevertheless, PLWH generated robust spike-specific T-cell responses following a post-vaccine breakthrough infection, reaching levels equivalent to those achieved after a fourth vaccine dose in SARS-CoV-2 infection-naive PLWH. Of note, receiving a fourth vaccine dose after having experienced a breakthrough SARS-CoV-2 infection enhanced immune responses even further, with CD8+ T-cell responses reaching levels significantly higher than those observed in SARS-CoV-2 infection-naive individuals after four doses. This finding is consistent with previous reports describing the superiority of “hybrid” immunity generated through a combination of vaccination and infection [33,34].

Our results are broadly consistent with other recent studies showing that vaccine-induced spike-specific T-cell responses are equivalent in PLWH and controls without HIV, particularly among PLWH receiving antiretroviral therapy [16,17,18,19,24,25]. We found no association between CD4+ T-cell responses and neutralizing antibodies after two or three vaccine doses. This suggests that sufficient T-cell help was provided to stimulate spike-specific B-cell responses despite a wide range of antigen-specific CD4+ T-cell frequencies induced by vaccination. While our results do not rule out potential dysfunction in vaccine-induced T cells among PLWH, our data suggest that the impact of T-cell impairment is likely to be modest after the second and third vaccine doses.

This study has some limitations. All PLWH received suppressive antiretroviral therapy and had generally preserved CD4+ T-cell counts, so our observations may not be generalizable to PLWH with low CD4+ T-cell counts or untreated HIV. Since this was an observational study, PLWH and control participants were not matched at enrolment. Though we included known variables in multivariable regression analyses, residual effects of variables that differed between groups (e.g., sex and vaccine regimen type) or other unknown factors could influence our results. We also lacked information on the concentration of mRNA-1273 third doses received (100 µg versus 50 µg), and therefore, we could not investigate this impact on immune responses. While the activation-induced marker assay provides a sensitive method to quantify spike-specific CD4+ and CD8+ T cells, we did not analyze T-cell subsets, such as follicular helper cells, nor T-cell functions, such as cytokine production or proliferation, which could reveal differences in the effector activity of vaccine- or infection-induced responses. Finally, since T-cell responses were measured using peptide pools spanning only the ancestral spike protein, we cannot comment on the relative dominance of individual T-cell epitopes nor potential changes in the distribution of T-cell responses following repeated vaccinations or a breakthrough infection.

## 5. Conclusions

Our study indicates that PLWH receiving suppressive antiretroviral therapy mount strong CD4+ and CD8+ T-cell responses to COVID-19 vaccines that are comparable to those observed in individuals without HIV. Repeated exposure to SARS-CoV-2 spike antigens, either through vaccination or infection, further enhanced T-cell responses among PLWH. HIV status was not associated with an increased risk of a post-vaccine breakthrough infection in our cohort, but older age was a significant predictor of protection against an infection. This counterintuitive outcome is consistent with lower SARS-CoV-2 anti-N seropositivity rates among older adults in Canada following the emergence of Omicron [35] and suggests that non-biological factors, such as maintenance of social distancing measures among older adults, contributed to the lower infection risk in this group. Together with the results of other recent studies, our results further underscore the benefits of COVID-19 vaccination in PLWH.

## Figures and Tables

**Figure 1 viruses-16-00661-f001:**
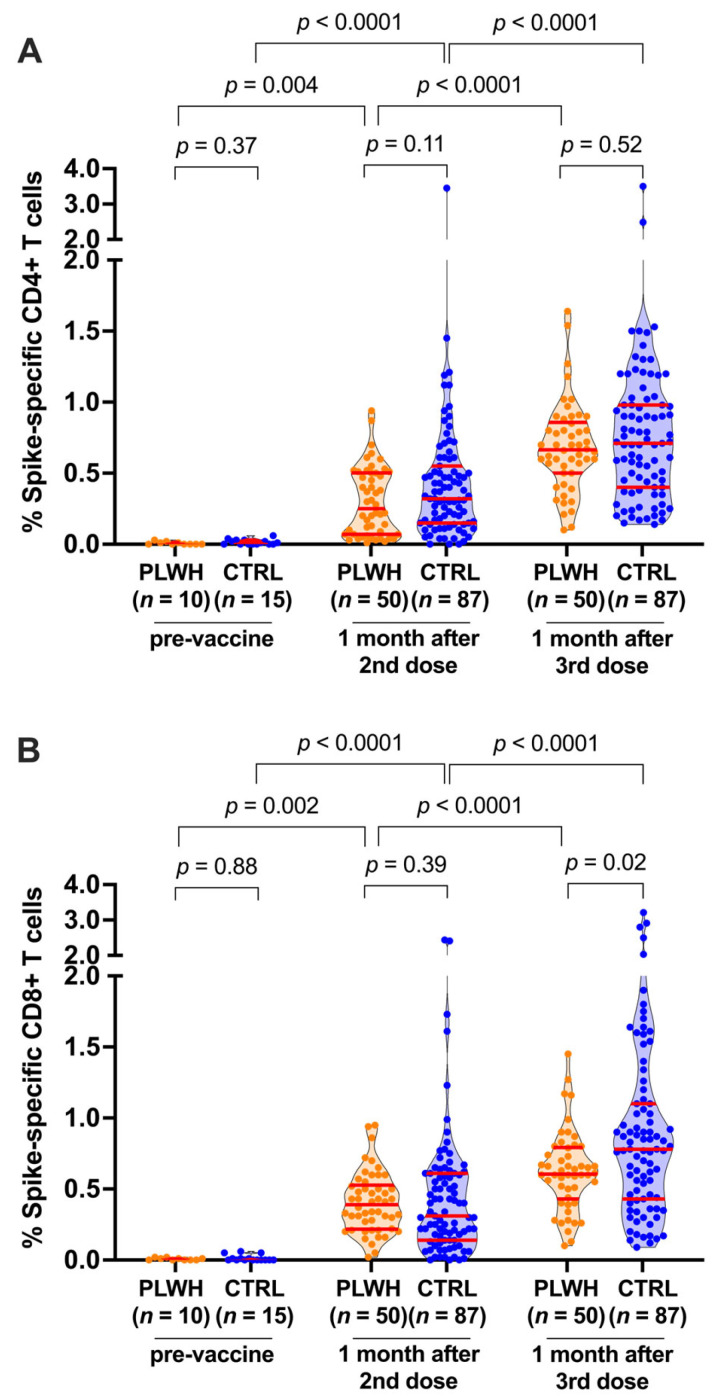
SARS-CoV-2 spike-specific T-cell responses after COVID-19 vaccination. (**A**) Spike-specific CD4+ T cells were quantified using the AIM assay (defined as OX40+/CD137+ cells) before and after two and three-dose COVID-19 vaccination. People living with HIV (PLWH) are in orange; controls without HIV (CTRL) are in blue. All participants are SARS-CoV-2 infection-naive. Red bars indicate the median and interquartile range. The Mann–Whitney U-test was used for between-group comparisons, and the Wilcoxon matched pairs test was used for longitudinal paired comparisons. *p*-values are not corrected for multiple comparisons. (**B**) Same as panel A, but for spike-specific CD8+ T cells quantified using the AIM assay (defined as CD69+/CD137+ cells).

**Figure 2 viruses-16-00661-f002:**
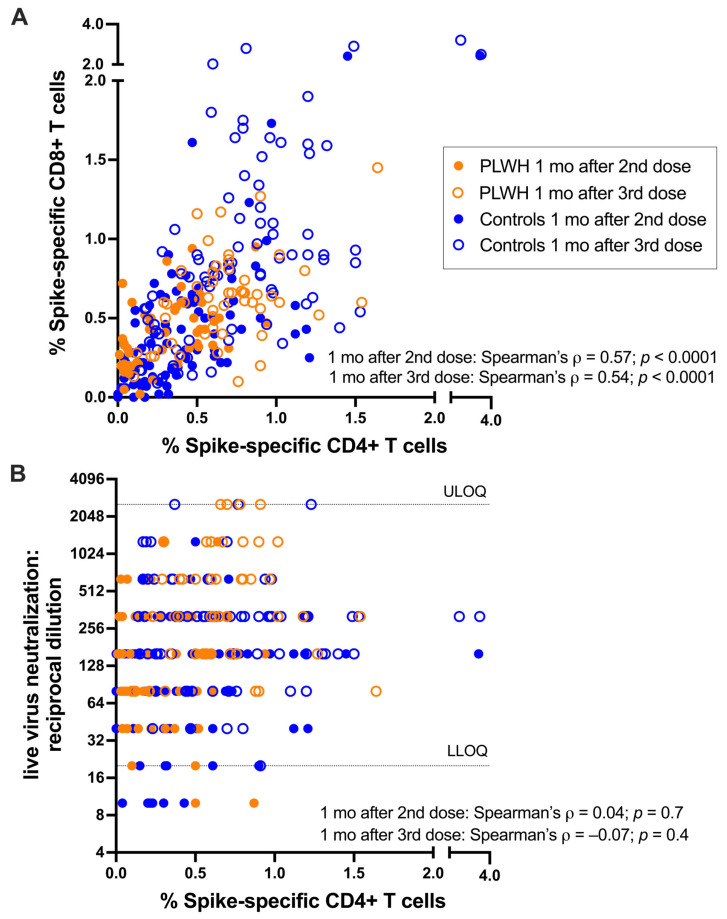
Associations between SARS-CoV-2 spike-specific cellular and humoral immune measures after COVID-19 vaccination. (**A**) Correlation between spike-specific CD4+ T cells (OX40+/CD137+) and CD8+ T cells (CD69+/CD137+) one month after the second dose (open circles) and one month after the third dose (closed circles) in people living with HIV (PLWH, in orange) and controls without HIV (in blue). (**B**) Correlation between spike-specific CD4+ T cells and SARS-CoV-2 neutralizing antibodies (reported previously [11,28]), measured one month after the second dose (open circles) and one month after the third dose (closed circles) in people living with HIV (PLWH, in orange) and controls without HIV (in blue). All participants are COVID-19-naive. The assay’s upper limit of quantification (ULOQ) and lower limit of quantification (ULOQ) are indicated.

**Figure 3 viruses-16-00661-f003:**
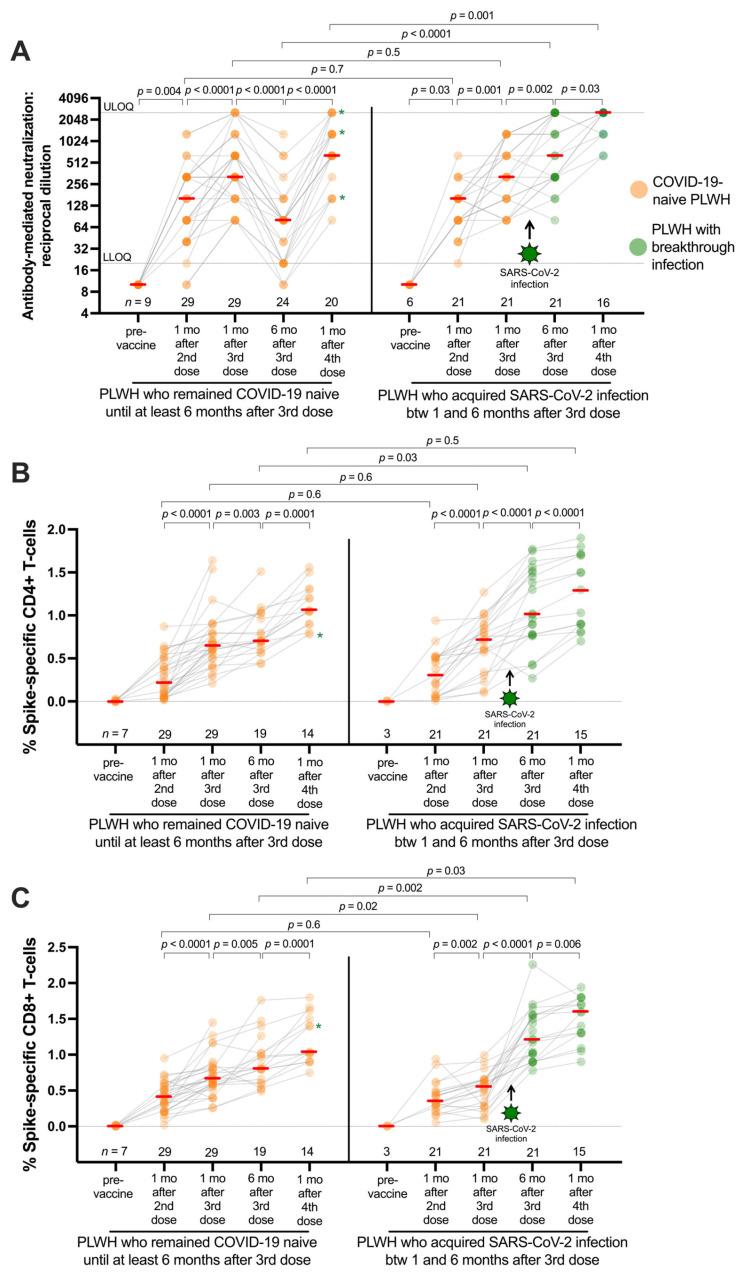
Longitudinal SARS-CoV-2 spike-specific immune responses in PLWH who remained naive or experienced their first SARS-CoV-2 infection after COVID-19 vaccination. (**A**) SARS-CoV-2 neutralizing antibody titers (reciprocal dilution) (reported previously [11,28]) in people living with HIV (PLWH) after two, three and four doses of COVID-19 vaccine. Grey lines connect data from the same individual. SARS-CoV-2-naive participants are displayed in orange. SARS-CoV-2 breakthrough participants after the third dose are displayed in green (starting at the point of seroconversion). Note that many antibody titer values overlap, resulting in darker-colored symbols. Asterisks denote naive participants who experienced SARS-CoV-2 breakthrough between six months post-dose third dose and one month post-fourth dose. Red bars indicate the median. The assay’s upper limit of quantification (ULOQ) and lower limit of quantification (LLOQ) are indicated. The Mann–Whitney U-test was used for between-group comparisons, and the Wilcoxon matched pairs test was used for longitudinal paired comparisons. *p*-values are not corrected for multiple comparisons. (**B**) Same as panel A, but for spike-specific CD4+ T-cell responses. (**C**) Same as panels A and B, but for spike-specific CD8+ T-cell responses. The Mann–Whitney U-test was used for between-group comparisons, and the Wilcoxon matched pairs test was used for longitudinal paired comparisons. *p*-values are not corrected for multiple comparisons.

**Figure 4 viruses-16-00661-f004:**
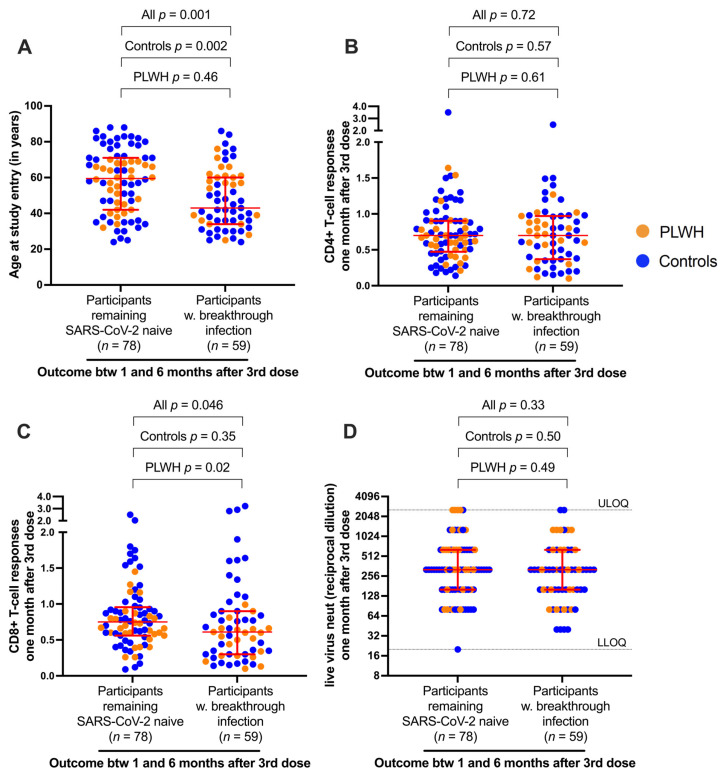
Analysis of correlates of protection against SARS-CoV-2 breakthrough infection after COVID-19 vaccination. (**A**) Participant ages are shown at study entry, stratified according to those who remained SARS-CoV-2-naive (*n* = 78) and those who experienced SARS-CoV-2 breakthrough infection between one and six months after receiving a third vaccine dose (*n* = 59). PLWH are in orange; controls are in blue. Red bars indicate median and interquartile ranges. (**B**) Same as panel A, except the frequencies of spike-specific CD4+ T cells one month after three vaccine doses are shown. (**C**) Same as panels A and B, except the frequencies of CD8+ T cells one month after three vaccine doses are shown. (**D**) Same as other panels, except SARS-CoV-2 neutralization activities (reported as reciprocal dilution titers) one month after three vaccine doses are shown. For all panels, the Mann–Whitney U-test was used to compare groups, using data from the entire cohort (all; *n* = 137), from controls only (*n* = 87), or from PLWH only (*n* = 50).

**Table 1 viruses-16-00661-t001:** Participant characteristics.

Characteristic	PLWH (*n* = 50)	Controls (*n* = 87)
Sociodemographic and health variables ^a^		
Age in years, median (IQR)	58 (42–65)	50 (35–72)
Female sex at birth, *n* (%)	6 (12%)	58 (67%)
Non-white ethnicity, *n* (%)	14 (28%)	37 (43%)
Number of chronic health conditions, median (IQR) ^b^	1 (0–1)	0 (0–1)
HIV plasma viral load (RNA copies per mL), median (IQR)	<50 (<50–<50)	N.A.
Recent CD4 cell count (per µL), median (IQR)	695 (468–983)	N.A.
Nadir CD4 cell count (per µL), median (IQR)	250 (140–490)	N.A.
Vaccine details		
Initial regimen ^c^		
mRNA only, *n* (%)	42 (84%)	84 (97%)
ChAdOx1-containing, *n* (%)	8 (16%)	3 (3%)
Third dose		
BNT162b2, *n* (%)	17 (34%)	34 (39%)
mRNA-1273, *n* (%)	33 (66%)	53 (61%)
Days between first and second doses, median (IQR)	59 (53–68)	89 (73–98)
Days between second and third doses, median (IQR)	182 (141–191)	197 (169–215)
Post-third-vaccine dose SARS-CoV-2 infections, *n* (%) ^d^	21 (42%)	38 (44%)

^a^ Sociodemographic, health, and vaccine data were collected by self-report and confirmed through medical records where available. CD4+ T-cell counts and HIV plasma viral loads were obtained from medical records. ^b^ Chronic conditions were defined as hypertension, diabetes, asthma, obesity, chronic diseases of lung, liver, kidney, heart or blood, cancer, and immunosuppression due to chronic conditions or medication. ^c^ Initial regimen refers to the first two doses. “mRNA only” refers to participants who received BNT162b2 and/or mRNA-1273 while “ChAdOx1-containing” refers to participants who received two ChAdOx1 doses, or one ChAdOx1 dose plus one mRNA dose. ^d^ SARS-CoV-2 infections that occurred between one and six months after the third vaccine dose.

## Data Availability

The datasets analyzed in this study are available from the corresponding author upon reasonable request. Data have also been deposited into the COVID-19 Immunology Task Force (CITF) data bank (https://portal.citf.mcgill.ca/).

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
