# Peer review of "T-Cell Responses to COVID-19 Vaccines and Breakthrough Infection in People Living with HIV Receiving Antiretroviral Therapy"

_viruses, 2024, doi:10.3390/v16050661_

Round 1
Reviewer 1 Report
Comments and Suggestions for Authors
Datwani and colleagues explore the immunogenicity of COVID-19 vaccines in PLWH, compared to HIV-uninfected subjects. In particular, they focus on cellular responses, a topic generally less studied than antibody responses.
They observe an overall intact cellular immunity, although slightly lower responses are detected in respect to CD8+ T cell responses. Notably, CD8+ T cell response are also identified in this paper as a correlate of protection against infection.
The data strongly support the results and the conclusions, and the Authors clearly approached the research questions exploiting different statistical methods that integrate each other.
The study has some limitations that the authors highlight, such as the difference in sex at birth between the study groups and the fact that PLWH may have received higher doses of COVID-19 vaccination or with different intervals between doses compared to Controls. This may potentially affect the outcomes. Indeed, several side-analysis suggest that the higher mRNA vaccine dose could boost T-cell responses ( see the correlation between CD8 responses and nadir CD4 count or the association with co-morbidities).
This aspect requires to be better highlighted in the discussion section. On top, this suggests that, even with the higher mRNA vaccine doses, PLWH have slightly lower CD8 responses.
Author Response
We thank the reviewer for their supportive comments. We agree that differences in vaccine dose between PLWH and controls may affect the outcomes of our study. Unfortunately, we have not been able to collect this information for most participants, so we can only speculate on the impact. Nevertheless, as requested by the reviewer, we have extended our discussion of this point (please see lines 380-385 in the revised manuscript).
Reviewer 2 Report
Comments and Suggestions for Authors
1. This is a study that compares the CD4+ and CD8+ T cell responses to COVID-19 vaccines in 50 people living with HIV (PLWH) and 87 controls without HIV. This is an observational study in which PLWH and control participants were not matched at enrollment. It is appreciated that the authors acknowledge other limitations in the design of this study (last paragraph of the discussion). However, the study is very well detailed, and the results are interesting.
2. The authors indicate that in the multivariable logistic regression analysis incorporating participant sociodemographic, vaccine and immune response variables, older age was the only significant predictor of protection against SARS-CoV-2 break through infection. However, there could be other factors not studied (nutritional situation, different intensity of T-cell immune response, etc.). Therefore, this interpretation should be modified in the Results section (lines 362-367) and in the Conclusion section (lines 431-433).
Author Response
We thank the reviewer for their interest in our work and thoughtful comments. We agree that additional factors that were not considered in our study could affect the reported outcomes. As suggested by the reviewer, we have modified the text to address this point (please see lines 369-373 in the revised manuscript).
Reviewer 3 Report
Comments and Suggestions for Authors
Dear Editor of Vaccines,
In the viruses-2949056 manuscript you kindly asked me to review, the authors report CD4+ and CD8+ spike-specific T-cell responses measured by AIM after COVID-19 infection and breakthrough infections and compared with the antibody neutralizing response. The antibody response to SARS-CoV-2 vaccination has been more widely studied than the T cell responses, although both arms of the adaptive immune response are necessary to control viral infection. Moreover, the development of a proper antibody response depends in many aspects of the CD4+ T cell response. Thus, in my opinion, the data reported by the authors is of interest to the reader of Viruses and all infectious disease researchers and certainly deserves to be published.
The manuscript is nicely written and needs only minor revisions to avoid some repetitions in the text, plus some additional information on the material and methods used to perform the flow cytometry analysis.
Comments:
Lines 23-27. This part fits better as material and methods and should be moved to this part of the abstract
Line 29. “All participants remained SARS-CoV-2 naïve until at”
should be changed for
“All participants remained SARS-CoV-2 infection naïve until at”
Line 59. There are other articles were the T cell immune response to SARS-CoV-2 vaccines was studied, for instance J Infect Dis. 2022 Nov 28;226(11):1913-1923
The authors should make an effort to make a more complete revision of previous studies
Lines 65-76. This paragraph is more appropriate for a conclusion section than for an introduction. Should be moved or shortened.
Line 102. How many OLP were used?, How many pools were used? How were they distributed?
Line 158. Sorry, I do not understand, an IQR between <50 and <50 is not just <50?
Lines 170-171. There are symbols missing? There are two “OBJ”
Line 179. Here and elsewhere, maybe µg is more used than mcg?
Lines 221-226. So, with this data showing that in PWH CD8 response is lower, can this be explained by CTL from PWH being more exhausted?
Line 231-236. Information on the criteria used to interpret the data should be in material and methods
Line 284. Is this new or already published data?
If it is new add the protocol to material and methods.
If it is already published add the reference herein
Table 1 has a red mark that should be eliminated.
Figure 1. Here and in other figures, the AIM used for CD4 and CD8 should be indicated in the figure. Also that the figures show SARS-CoV-2-specific responses.
Figure 3. Increase the resolution of this figure
Author Response
We thank the reviewer for their thoughtful comments. We have made the following edits to our manuscript (in Red text below).
Reviewer Comments:
Lines 23-27. This part fits better as material and methods and should be moved to this part of the abstract –
We have moved this line in the abstract and also removed the abstract sub-headers.
Line 29. “All participants remained SARS-CoV-2 naïve until at” should be changed for “All participants remained SARS-CoV-2 infection naïve until at”
We have made this change throughout the manuscript.
Line 59. There are other articles were the T cell immune response to SARS-CoV-2 vaccines was studied, for instance J Infect Dis. 2022 Nov 28;226(11):1913-1923. The authors should make an effort to make a more complete revision of previous studies
We now reference a larger number of relevant publications related to vaccine-induced T cell responses in PLWH, including the JID study mentioned by the reviewer.
Lines 65-76. This paragraph is more appropriate for a conclusion section than for an introduction. Should be moved or shortened.
We have shortened this paragraph.
Line 102. How many OLP were used?, How many pools were used? How were they distributed?
We apologize for the omission. We include additional details of the peptide set, which included 315 individual peptides and was obtained from a commercial source (Miltenyi Biotec).
Line 158. Sorry, I do not understand, an IQR between <50 and <50 is not just <50?
The IQR reflects the 25% and 75% percentile of the distribution, which were also below 50 copies/mL. Since this information is also reported in Table 1, we have removed the IQR values from the text to avoid confusion.
Lines 170-171. There are symbols missing? There are two “OBJ”
We apologize, this appears to be a typo in the file. No symbols or text are missing.
Line 179. Here and elsewhere, maybe µg is more used than mcg?
We have replaced mcg with µg, as suggested.
Lines 221-226. So, with this data showing that in PWH CD8 response is lower, can this be explained by CTL from PWH being more exhausted?
The reviewer is correct. This result is consistent with T cell exhaustion that is observed in many PLWH despite receiving therapy. We have added this potential explanation to our results (see lines 225-226).
Line 231-236. Information on the criteria used to interpret the data should be in material and methods
We thank the reviewer for this suggestion. We have removed this text from the Results, since it is already discussed in the Methods.
Line 284. Is this new or already published data? If it is new add the protocol to material and methods. If it is already published add the reference herein
The neutralization data was reported previously – these papers are cited on line 284 in the revised manuscript and in the figure legends.
Table 1 has a red mark that should be eliminated.
This appears to be due to due to an error check conducted by Microsoft Word – we will remove it if necessary at the proof stage.
Figure 1. Here and in other figures, the AIM used for CD4 and CD8 should be indicated in the figure. Also that the figures show SARS-CoV-2-specific responses.
We apologize for omitting this important information. We have now added these details to all figure legends.
Figure 3. Increase the resolution of this figure
We have submitted high-resolution images of this and other figures for publication.